# The Role of Arthroscopy in Contemporary Glenoid Fossa Fracture Fixation

**DOI:** 10.3390/diagnostics14090908

**Published:** 2024-04-26

**Authors:** Byron Chalidis, Polychronis P. Papadopoulos, Pericles Papadopoulos, Charalampos Pitsilos

**Affiliations:** 11st Orthopaedic Department, Aristotle University of Thessaloniki, 57010 Thessaloniki, Greece; 22nd Orthopaedic Department, Aristotle University of Thessaloniki, 54635 Thessaloniki, Greece; dr.polychronis.papadopoulos@gmail.com (P.P.P.); perpap@otenet.gr (P.P.); chpitsilos@outlook.com (C.P.)

**Keywords:** glenoid fossa, glenoid fracture, intra-articular fracture, shoulder arthroscopy, fracture fixation

## Abstract

Glenoid fossa fractures are rare injuries accounting for 10 to 29% of all intra-articular scapula fractures. They are usually the result of high-energy trauma, and concomitant injuries are not uncommon. Patients with glenoid fractures are admitted with shoulder pain and restricted range of motion. Although shoulder plain radiographs could establish the diagnosis, a computed tomography scan is necessary to adequately define the fracture pattern and characteristics. The most commonly used classification system is that of Ideberg (modified by Goss), which includes five glenoid fossa fracture types according to the location, extension, and complexity of the lesion. Articular surface displacement and step-off are the most important factors that should be taken under consideration when deciding for conservative or surgical management. Operative treatment includes open reduction and internal fixation through a posterior or anterior approach depending on fracture morphology and displacement. However, open surgical techniques are related to extensive soft-tissue disruption, risk of neurovascular injury, and inadequate exposure of the entire glenoid cavity. Introduction of arthroscopy could facilitate better visualization of the glenoid articular surface and improved fracture reduction. However, it is a technically demanding procedure with many challenges and pitfalls. The aim of this review is to summarize the current evidence regarding the treatment of glenoid fossa fractures and present the beneficial effect of arthroscopy in improving the quality of fracture fixation and overall functional outcomes.

## 1. Introduction

The glenohumeral joint is a non-conforming joint that consists of a concave and shallow glenoid and a spherical humeral head [1]. This ball-and-socket structure allows an extreme range of motion in multiple planes, which predisposes to joint instability [2]. The latter is more frequently encountered when a traumatic shoulder dislocation event is complicated with a glenoid rim fracture [3]. Avulsion of the anterior glenoid rim is the most common type of all scapula fractures [4]. Glenoid fossa fractures are less common, with a relevant rate of 10 to 29% of all scapula fractures [5,6]. This incidence makes this injury extremely rare, considering that scapula fractures represent only 1% of all body fractures and 3% of shoulder area fractures [7].

Glenoid fossa fractures are usually the result of a high-energy trauma [8]. Thus, thorough examination of the patient is essential to exclude possible concomitant injuries [9]. The most common associated injury sites are the chest, including rib or clavicle fractures and pneumothorax, the head, and the spine [10]. In case of shoulder pain, swelling, tenderness, and deformity, plain radiographs of the affected shoulder should follow the standard trauma imaging examination [11,12]. However, shoulder X-rays are not always sensitive and accurate to define a glenoid fracture and, therefore, a computed tomography (CT) scan is always necessary to precisely describe the fracture morphology and reveal any occult injuries [13]. Based on the imaging studies, these fractures can be classified into different types, with the most widely used classification system being that of Ideberg, modified by Goss [14].

In contrast to most extra-articular fractures of the scapula body, neck, or spine, which can be treated conservatively with satisfactory reported outcomes, intra-articular glenoid fractures often necessitate surgical treatment to restore articular congruity and glenohumeral joint stability and facilitate early functional rehabilitation [15,16,17]. Non-operative treatment of displaced glenoid fractures has been associated with chronic pain, stiffness, malunion, development of early glenohumeral osteoarthritis, and poor outcomes [18,19]. 

Open reduction and internal fixation through anterior or posterior approaches have been the standard of surgical care of displaced glenoid fractures [20,21,22]. However, the open approaches require extensive soft-tissue dissection and pose the risk of neurovascular structures’ injury. Besides, a single approach could not always allow sufficient and wide exposure of the entire glenoid cavity and may be correlated with poor visualization and inadequate fracture reduction [23,24,25,26]. 

To overcome these disadvantages of open surgery and improve injury visualization, arthroscopy-assisted fracture reduction and internal fixation were introduced initially for the treatment of anterior glenoid fractures associated with joint instability [27,28,29]. Since then, arthroscopically assisted fixation techniques have been applied to address more complicated intra-articular scapular fractures, including Ideberg type II to VI fractures [30]. The purpose of this review is to summarize the current evidence regarding the optimal diagnostic strategy and treatment process of glenoid fossa fractures and discuss the potential benefit of arthroscopy in improving the quality of fracture reduction and fixation, as well as the functional outcome.

## 2. Anatomy

The glenoid is typically a pear-shaped cavity (48.6%), with the superior–inferior diameter being larger than the anteroposterior and the lower part being wider than the upper one [31]. It has also been described as inverted comma-shaped (34%) or oval-shaped (17.4%) [32,33]. 

It has a native retroversion with respect to the scapula axis and the relevant value is highly correlated with hand dominance [34]. Additionally, the glenoid has superior inclination related to the transverse axis, which prevents inferior subluxation of the humeral head [35]. 

The glenohumeral joint is described as “ball and socket”, as it consists of the concave glenoid and the convex humeral head [36]. However, only 25% of the large humeral head is articulated with the glenoid fossa [37]. The depth of the fossa is shallow as a result of the bone morphology and the relatively increased thickness of the overlying articular cartilage [38]. Therefore, the surrounding fibrocartilaginous labrum is very valuable because it deepens the cavity and increases the contact surface between the glenoid and the humeral head, and subsequently the overall glenohumeral joint stability [39]. Apart from the labrum, other static stabilizers of the joint are the joint capsule, the negative intra-articular pressure, and the local ligaments, which prevent humerus luxation [40]. The dynamic stabilizing structures of the glenohumeral joint include the rotator cuff and the periscapular muscles, as well as the long heads of the biceps and the triceps tendons [41]. 

The last two tendons play a significant role in fracture morphology and characteristics. The long head of the biceps tendon is attached to various bony landmarks of the supraglenoid tubercle of the scapula and superior labrum [42]. On the other hand, the long head of triceps muscle originates from the infra-glenoid tubercle of the scapula [43]. The pull of these two tendon structures, along with the traction of the glenohumeral ligaments and joint capsule, present the main deformity forces of glenoid fossa fractures [18,44]. 

## 3. Diagnosis

### 3.1. Mechanism of Injury and Clinical Findings

Glenoid fossa fractures mainly occur after high-energy injuries [23]. Motor vehicle accidents and falls from height are the most common causes [45]. In these cases, a direct axial force results in impaction of the glenoid articular surface by the humeral head [13]. This typically creates a transverse fracture along the direction of the force trajectory [18]. The high-impact trauma, which is usually responsible for the glenoid fossa fractures, may also cause concomitant injuries and fractures to the clavicle, humerus, chest wall, head, vertebrae, pelvis, or brachial plexus, necessitating further investigation and treatment [46,47,48,49].

Clinical examination of the shoulder region after injury requires the thorough and independent evaluation of the glenohumeral, sternoclavicular, acromioclavicular, and scapulothoracic joints [50]. The active and passive range of motion, the muscle strength, and the neurovascular status should be accessed, particularly after high-velocity injuries [51]. Patients with glenoid fossa fractures may demonstrate restricted range of motion and pain that are exacerbated with abduction and rotational movements [12]. In addition, local ecchymosis and tenderness to palpation of the shoulder may also be apparent [52]. 

### 3.2. Imaging

The investigation of patients with suspicion of intra-articular scapula fractures has traditionally included the standard shoulder trauma radiographs, such as the true anteroposterior (Grashey) view, the axillary view, and the axial scapula (“Y”) view [13,53,54]. These X-ray images are useful to evaluate not only the glenohumeral joint injuries but also any associated humerus fractures, clavicle or rib fractures, acromioclavicular separation, and scapulothoracic dissociation [12].

The conventional CT scanning is currently recommended in all cases with intra-articular scapula fractures and complex shoulder injuries. It could also be applied for revealing any occult fractures, especially when severe shoulder pain is correlated with normal plain radiographs [13]. Interpretation of conventional X-rays in the children and adolescent population should be performed with caution, as the normal physis of the secondary ossification center that arises from the inferior part of the glenoid and forms three-fourths of the glenoid cavity may be misdiagnosed for fracture of the glenoid [55]. In this scenario, CT screening is necessary to confirm or rule out a glenoid fracture [55,56]. 

The three-dimensional (3D) CT reconstruction with the potential of multiple angle views may illustrate the fracture pattern more precisely and add more information useful to prepare the preoperative plan of the surgically treated glenoid fractures [57]. Recently, a sequence of magnetic resonance imaging (MRI), the 3D T1-weighted gradient-echo sequence or high-resolution CT-like MRI, has been used for the evaluation of osseous pathology of the shoulder joint. It has shown high accuracy in the diagnosis of glenoid fractures, and it has displayed comparable results to the CT scan [58]. Conventional MRI can also be used to assess the integrity of the rotator cuff tendons and brachial plexus, particularly when neurologic examination could not be utilized in unconscious and mechanically ventilated patients [59]. 

## 4. Classification

Ideberg et al. [60] were the first to attempt to classify the glenoid fractures, and they categorized them into five types. Goss [14] later modified the original Ideberg classification, and this modification is the most commonly used system for classifying intra-articular scapula fractures. It includes six types, five of which refer to glenoid fossa fractures (Figure 1). Type I is a glenoid rim fracture, while subtype Ia includes fractures of the anterior rim and subtype Ib fractures of the posterior rim. Type II is an inferior glenoid fracture involving part of the neck. In type III injuries, the fracture line extends through the base of the coracoid process. In type IV fractures, a horizontal fracture line runs inferior to the spine of the scapula involving both the scapula neck and body. Type V is a combination of type II, III, and IV fractures and is divided into three subgroups, based on the fracture pattern. Finally, severe comminuted fractures of the glenoid articular surface are described as type VI.

The “Arbeitsgemeinschaft für Osteosynthesefragen” (AO) Foundation and the Orthopedic Trauma Association (OTA) described a different classification for glenoid fractures, including five types, three of which referred to glenoid fossa fractures [61] (Figure 2). A simple vertical or oblique fracture, which creates an anterior or posterior free fragment, presents a 14F1.1 and 14F1.2 type, respectively. In type 14F1.3 fractures, there is a simple transverse or short oblique fracture. Type 14F2.1 fractures have a multi-fragmentary pattern with at least three fragments, and type 14F2.2 fractures correspond to multi-fragmentary fractures with associated central dislocation. The AO/OTA classification system seems to have a better inter- and intra-observer reliability than the modified Ideberg classification system [62]. 

Euler and Rüedi [62] also classified the scapula fractures into five main categories, based on the most commonly involved anatomical location. The intra-articular glenoid fractures are included in the type D category. There are three subgroups within this type: D1 includes rim fractures, D2 glenoid fossa fractures, and D3 glenoid cavity fractures combined with glenoid neck or body fractures. Type D2 fractures are further divided into subgroup D2a, where there is an inferior glenoid fragment, subgroup D2b that refers to a transverse intra-articular fracture, subgroup D2c, where there is a separate coracoglenoid block fragment, and subgroup D2d, which includes multiple glenoid bone fragments. This classification also has a lower inter-observer reliability and confidence than the AO/OTA classification for the intra-articular glenoid fractures [63]. 

## 5. Management

There is controversy regarding the most appropriate treatment option for the treatment of glenoid fossa fractures [52]. The decision depends on the degree of displacement, the presence of intra-articular gap and step-off, the fragment size, and the occurrence of glenohumeral joint instability [64,65]. 

### 5.1. Conservative Treatment

Conservative treatment of intra-articular glenoid fractures includes a small period of sling immobilization of the shoulder joint, followed by early range of motion and rehabilitation [66]. It is indicated in cases of minimal displacement with gap or step-off of up to 3 to 4 mm and involvement of less than one-third of the articular surface [67]. Königshausen et al. [68], in their case series of 24 patients with conservatively treated glenoid fossa fractures, found that a persistent fracture gap of more than 5 mm was correlated to non-union, pain, stiffness, and a poor functional outcome. 

### 5.2. Open Reduction and Internal Fixation

The standard of treatment for displaced intra-articular glenoid fractures is operative management with open reduction and internal fixation [18]. Although surgery is not necessary to take place on an urgent basis, it is preferrable to be performed within the first three weeks after the injury [69]. Any further delay maximizes the difficulties of fracture reduction and the risks of cartilage degeneration, joint stiffness, and post-traumatic arthritis [68]. Fixation can be achieved by using screws (cannulated or not) only, or plates and screws [70,71].

According to fracture location and morphology, two main approaches are generally used: the dorsal approach for posterior glenoid fractures and the deltopectoral approach for fractures with anterior involvement [72,73]. The Judet (posterior) approach allows exposure of the body and articular part of the scapula but requires a large skin incision and detachment of the deltoid muscle [20]. Many modifications of the Judet approach have been described, utilizing a more centralized incision, less soft-tissue retraction, and minimal muscular stripping [74,75,76,77]. The deltopectoral approach allows easy access to the anterior rim and body of the glenoid and can be used for almost any shoulder fracture fixation [78,79]. The axillary approach has also been described as a feasible surgical option for Ideberg type II fractures, as it is associated with less surgical trauma and bleeding and good visualization of the inferior anterior aspect of the scapular glenoid [80]. 

Ao et al. [25], in a cadaveric study, compared the extent of the glenoid exposure between the deltopectoral and the Judet approaches. They found that neither of them could effectively expose the glenoid articular surface and provide successful open reduction and internal fixation of complex intra-articular fractures. Thus, more extensive approaches have been proposed for the surgical treatment of Ideberg V or VI fractures [81]. A combination of anterior and posterior approaches was used by Mayo et al. [82] for fixation of comminuted glenoid fossa fractures. More recently, Hong et al. [83] presented the “deltoid takedown” approach technique, which provided extensive exposure of the glenoid cavity.

The open reduction and internal fixation of glenoid fossa fractures may be complicated in approximately 9% to 18% of operated cases according to the extent of the applied surgical exposure [12]. Heterotopic ossification, intra-articular screw placement, postoperative joint stiffness, deep infection, non-union, and suprascapular nerve palsy with secondary infraspinatus muscle denervation may be encountered after operative treatment of displaced glenoid fractures [84,85,86]. Therefore, meticulous and careful dissection of deeper structures and layers must be carried out, and excessive traction should be avoided throughout the whole surgical procedure.

## 6. The Role of Arthroscopy

The arthroscopic approach for the treatment of glenoid cavity fractures yields many advantages, including proper visualization of the articular surface, minimal soft-tissue dissection, less disruption of blood supply and blood loss, a lower infection rate, improved cosmetics, and more rapid rehabilitation [87,88]. In contrast to open surgery, articular penetration and damage from the applied implants and screws are prevented due to direct vision of the glenoid fossa [89]. Additionally, simultaneous tendon, ligament, and capsular injuries can be diagnosed and effectively addressed at the same time as fracture fixation [90]. However, arthroscopy-assisted fixation of glenoid fossa fractures is a technically challenging procedure, and its application in complex scapula fractures may be very difficult or near impossible [28,88] (Table 1).

The role of arthroscopy in the treatment of Ideberg Ia (bony Bankart) fractures has been well established in the literature over the last two decades [91]. These fractures, which are most commonly associated with anterior shoulder instability, are fixed almost exclusively arthroscopically using different methods of fixation [92]. According to the size and configuration of the fractures and the surgeon’s preference, absorbable or non-absorbable suture anchors, screws, and trans-osseous sutures can be used for stabilization of the anterior glenoid rim [93,94,95]. However, in case of extensive, comminuted, or complicated intra-articular scapula fractures, an open approach is preferred to manage fracture displacement and address any concomitant neurovascular injuries [19]. 

As glenoid fossa fractures are intra-articular fractures, anatomic reduction of the articular surface is of the utmost importance [96]. Arthroscopy allows direct visualization of the glenoid cavity and assessment of joint cartilage and stability [97] (Figure 3).

Percutaneous application of elevators or Kirschner wires can be used to disengage, mobilize, and reduce the bone fragments in order to restore the articular congruity [97,98]. Afterward, the fractures are provisionally stabilized with the use of a reduction clamp or with the advancement of Kirschner wires and guide pins of cannulated screws under fluoroscopy [28,99,100]. The final fixation is typically achieved with percutaneously placed or cannulated screws [28,101]. Postoperative CT scans can be performed to evaluate the quality of fracture reduction and check the screw position and length [102] (Figure 4).

A review of the literature on studies about the arthroscopy-assisted fixation of glenoid fossa fractures was conducted, and the results are summarized in Table 2, Table 3 and Table 4. The inclusion criteria comprised of clinical studies or case reports with a minimum follow-up of six months that were published in English language journals and provided details about the fracture union, postoperative shoulder range of motion, and functional outcome.

### 6.1. Ideberg Type III Fractures

Arthroscopic reduction and percutaneous fixation have also been successfully performed in Ideberg type III glenoid fractures [28,88,98,102,103,104,105]. In most cases, the beach-chair position is selected, and the C-arm fluoroscope is placed in ipsilateral or contralateral side of the affected limb [88,102]. However, some surgeons prefer the lateral decubitus position [98]. After arthroscopic debridement, the superior fragment is usually manipulated and reduced using Kirschner wires [102]. As the coracoid process is in continuity with the displaced superior part of the articular surface, a serrated reduction clamp, or “lobster jaw”, can be used to grasp the process and obtain satisfactory fracture reduction [28,88,105]. Other instruments, such as arthroscopic probes or elevators, could be further applied to dis-impact and manipulate the fracture fragments [28,98]. 

When adequate reduction of the Ideberg type III fractures is confirmed, the wires used for manipulation of the superior glenoid fossa are advanced to the inferior part. Cannulated screws are subsequently introduced to stabilize and compress the fracture site [102]. Supplementary materials could also be used during arthroscopy to achieve the optimal position of the guide wire. Zbili et al. [103] introduced the Arthrex Tightrope (Arthrex Co., Ltd., Naples, Florida, USA) device ancillary through the Neviaser and the anterior portals to introduce the guide pin for the cannulated screw. Similarly, Yang et al. [104] inserted the long arm of the anterior cruciate ligament jig through the anterior portal with the tip at the inferior rim of the glenoid to determine the trajectory of the guide wire. The authors suggested to avoid instrument insertion close to the inferior pole of the glenoid fossa due to the high risk of damaging the axillary nerve. 

Various arthroscopic portals and landmarks have been used to aid fixation of glenoid fractures. The Neviaser portal is commonly applied for internal fixation of simple transverse glenoid fractures, as it allows perpendicular insertion of the screw to the fracture plane [88,103,104,105]. However, as the introduction of the screw through this portal may injure the suprascapular nerve and the supraspinatus tendon, some authors have suggested to avoid it for fracture fixation [9]. Marsland et al. [102] inserted two guide wires for the cannulated screws percutaneously, just anterior to the clavicle and medial to the acromioclavicular joint. Wafaisade et al. [28] used the high anteromedial portal for introduction of a Kirschner wire at the 12:30 o’clock position of the superior glenoid rim. 

Cannulated screws are the most frequently used devices for glenoid fossa fractures’ fixation [88,104]. However, the number of screws and the length of the thread depend mainly on surgeon preference. Usually, one or two screws are inserted under direct arthroscopic vision and radiological control for Ideberg type III fractures [103,105]. In simple transverse fractures, even one screw can achieve adequate stability [104]. The second screw is commonly applied in case of intra-articular comminution, as it facilitates a more stable fixation and allows earlier active motion [102]. In these cases, the screws are tightened alternately to minimize the rotational torque and achieve symmetrical compression of the bone fragments [105]. Additionally, partially threaded cannulated screws are most commonly preferred, as they can provide sliding compression of the fracture site with minimal soft-tissue dissection [88,98]. Compression of the glenoid fossa fracture could also be achieved with fully threaded screws if they are used as lag screws by over-drilling the near cortex [103]. The final position of fixation screws is thoroughly checked radiologically using the C-arm so as not to protrude too far from the inferior glenoid rim [105]. The arthroscopic imaging offers a global view of the glenoid fossa and confirms that the screw does not violate the articular surface [104].

Postoperatively, the shoulder joint is protected using a sling and immobilized for a period of approximately four to six weeks [102,105]. However, passive range of motion within pain tolerance may be initiated immediately according to the fracture fixation stability [104]. Bonczek et al. [105] allowed active mobilization from the third week after arthroscopic-assisted fixation of an Ideberg type III glenoid fracture using two screws, as they considered that this fixation was stable enough to withstand normal mechanical forces. However, and in most operated cases, active-assisted or active range of motion exercises are reasonable to be delayed until the sixth week after surgery [103].

Radiological fracture union is anticipated to be seen in plain radiographs between the second and third months after fixation [104]. The shoulder mobility without any major discomfort may return to functional independence by the third month, while full range of motion and sufficient muscle strength are usually achieved at six months [103]. A full return to pre-injury occupation and daily activities is expected within the first six months, and even heavy manual duties could be effectively performed at this time point [102,104]. 

### 6.2. Other Types of Glenoid Fossa Fractures

The arthroscopic-assisted fixation of the other types of intra-articular scapula fractures has not been extensively studied compared to the Ideberg type III fractures. Zhang et al. [106] reported four patients with concomitant Ideberg II glenoid and greater tuberosity fractures who were treated surgically with an arthroscopically assisted fixation procedure. In their surgical technique, the fragment of the inferior glenoid was fixed using two, three, or four suture anchors and a cannulated screw. The greater tuberosity was re-attached using a trans-osseous equivalent anchor technique with or without additional cannulated screws. All fractures were healed, and no complication was observed in a mean period of 38.2 months follow-up. 

The surgical treatment of Ideberg type IV glenoid fractures is similar to that for the type III variant pattern, as the orientation of the fracture line is also transverse [102,103]. Marshland et al. [102] reported a case of a male patient with a “T-shaped” fracture, where the transverse part showed a displacement of approximately 5 mm and the vertical one was not displaced. Twenty-one days post-injury, the patient underwent arthroscopy-assisted percutaneous fixation with a single screw that was inserted in a superior to inferior direction, resulting in a reduced and stable construct. At one-year follow-up, full range of motion was regained, and the total Constant–Murley shoulder score was 94 out of 100.

In type V fractures, the glenoid articular surface discontinuity is accompanied by a scapula body fracture [14]. The arthroscopic technique can address only the intra-articular part; however, the concomitant body fracture does not necessarily require operative treatment [107]. In rare surgical cases, scapula body osteosynthesis may be performed through a different approach [101]. Tuman et al. [107] described the arthroscopic-assisted fixation of the type V pattern in the lateral decubitus position. After arthroscopic debridement and standard diagnostic arthroscopy, a probe or a Bankart elevator was used through the anterior portal for manipulation of the bone fragments and subsequent fracture reduction. In order to allow access to the inferior glenoid, an additional posteroinferior portal was created. A guide wire was inserted through the cannula of the posteroinferior portal in a posteroinferior or anterosuperior direction, and a second guide wire was introduced percutaneously, adjacent to the first one, to maintain the reduction. Fracture compression and fixation were achieved using a partially threaded cannulated screw that was placed over the cannula. The integrity of the articular surface was confirmed with direct arthroscopic visualization, and the position and length of the screw were assessed with intraoperative fluoroscopy. A slightly different approach was described by Ulusoy [101] for the surgical treatment of an Ideberg Vb glenoid fracture in a male laborer patient. The patient was placed in a beach-chair position and, after arthroscopic hematoma removal, a Kirschner wire was introduced through the Neviaser portal to aid fracture reduction. Two guide pins were introduced from the same portal parallel to the Kirschner wire and, after fluoroscopic confirmation of their proper position, two partially threaded screws were inserted to stabilize and compress the fracture site. Postoperatively, a sling was used for six weeks, and passive range of motion exercises were initiated from the first day. At six months, the Disabilities of the Arm, Shoulder, and Hand questionnaire score was 2 of 100, and the University of California at Los Angeles shoulder score was 33 of 35.

The operative treatment of Ideberg type VI glenoid fractures is challenging, as their complexity may obscure successful and anatomic fracture reduction [5]. Many different approaches have been described for the arthroscopy-assisted fixation of this fracture pattern [99,100,108]. Gigante et al. [100] described the arthroscopic fixation of a “Y-shaped” glenoid fracture using two percutaneously placed Kirschner wires. After manipulation and reduction of the superior and the posterior bone fragments with a blunt periosteal elevator, fracture fixation was utilized with a wire that was introduced from the anterior to posterior direction. A second wire was also inserted from posterior to anterior for fixation of the remaining bone fragment. The Kirschner wires were left protruding through the skin and removed six weeks postoperatively. Bauer et al. [108] used trans-osseous sutures for the fixation of a type VI fracture with displaced anterior and posterior glenoid fragments. In their technique, non-absorbable sutures were initially passed through the anterior fragment for accomplishing fracture manipulation and reduction. Then, a long drill made a hole in an anterosuperior to posteroinferior direction, and the ends of the sutures were passed through the hole. Finally, the anterior and posterior fragments were reduced using a probe, and the suture ends were tied on the fascia of the infraspinatus tendon. Qu et al. [99] published a case series of eleven patients with comminuted glenoid fracture treated arthroscopically using a combination of cannulated screws and suture anchors. Two to three cannulated screws were inserted perpendicular to the fracture line between the major bone fragments, while smaller parts were fixed with suture anchors. A statistically significant improvement of the range of motion, the American Shoulder and Elbow Surgeon assessment, and the Rowe score was found between the preoperative and the last postoperative assessment at a mean period of 21 months. 

A rare case of a middle-aged laborer with glenoid fossa fracture non-union after conservative treatment, managed with arthroscopy-assisted grafting and fixation, was reported by Sears and Lazarus [109]. Fracture non-union was proved with imaging studies after a period of five months from the injury. Arthroscopy-assisted percutaneous fixation and bone grafting with cancellous allograft were effective in achieving osseous union, which was confirmed by a computed tomography scan at four months postoperatively. 

### 6.3. Comparison of Arthroscopy-Assisted Fixation and Open Reduction and Internal Fixation of Glenoid Fossa Fractures

A comparative study between open reduction and internal fixation through the Judet approach and arthroscopy-assisted fixation of intra-articular glenoid fractures was conducted by Lin et al. [30]. The authors included Ideberg type II through VI fractures. Twenty-four patients were operated on with open surgery and twenty patients underwent percutaneous fixation via the arthroscopic technique. At a mean follow-up of fifteen months, the range of motion was similar in both groups, but the Visual Analogue Score was significantly better in the arthroscopy group. The Constant and the Disabilities of the Arm, Shoulder, and Hand questionnaire scores were worst in the open approach group, but not statistically significant. Interestingly, and apart from bone injuries, a number of associated soft-tissue lesions were also identified and consequently addressed during the arthroscopic procedure. 

## 7. Conclusions

Shoulder arthroscopy could play a significant role in assisting the management and fixation of glenoid fossa fractures. Direct visualization of the intra-articular fracture of the glenoid offers better fracture control and safe introduction of percutaneous screws, avoiding penetration of the articular surface and implants’ malposition. In addition, the advantages of reduced postoperative pain due to less soft-tissue damage and potential acceleration of the rehabilitation process, compared to conventional open procedures, make arthroscopy-assisted fixation a viable and effective option for the operative treatment of glenoid fossa fractures, with promising functional results.

## Figures and Tables

**Figure 1 diagnostics-14-00908-f001:**
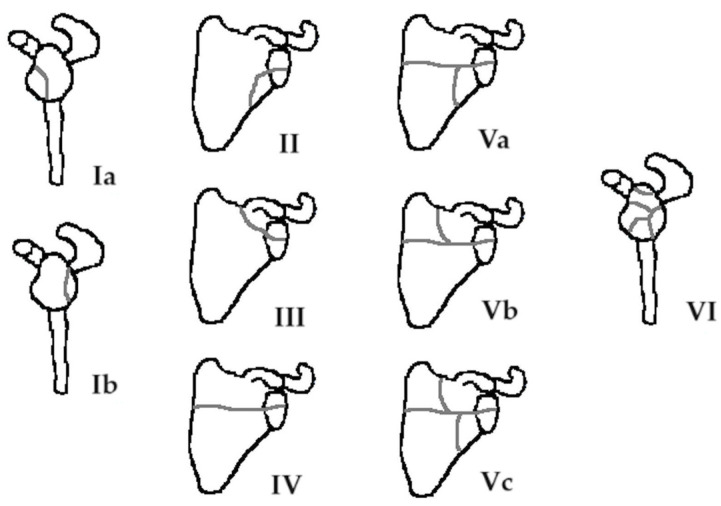
The Ideberg–Goss classification of glenoid fractures [14].

**Figure 2 diagnostics-14-00908-f002:**
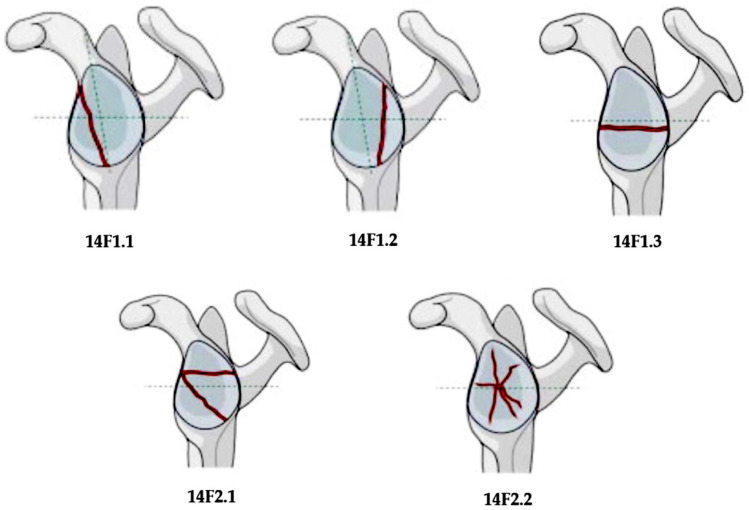
AO/OTA classification of glenoid fractures.

**Figure 3 diagnostics-14-00908-f003:**
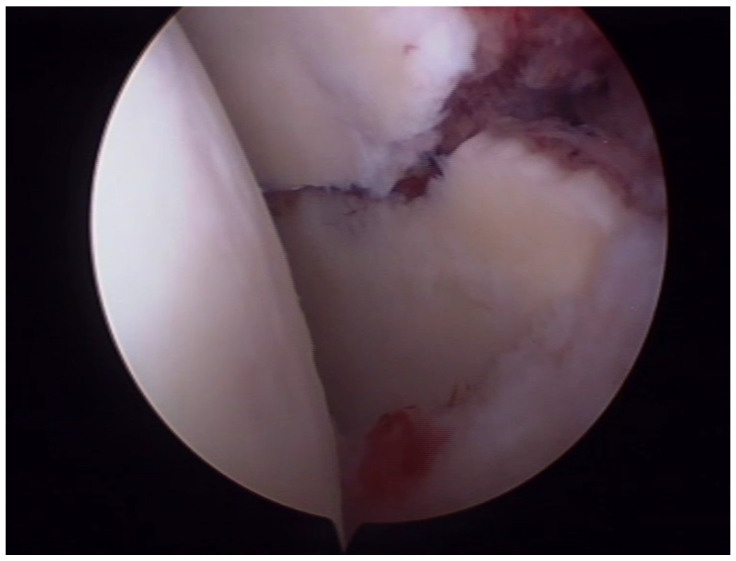
Arthroscopic image of a glenoid fossa fracture.

**Figure 4 diagnostics-14-00908-f004:**
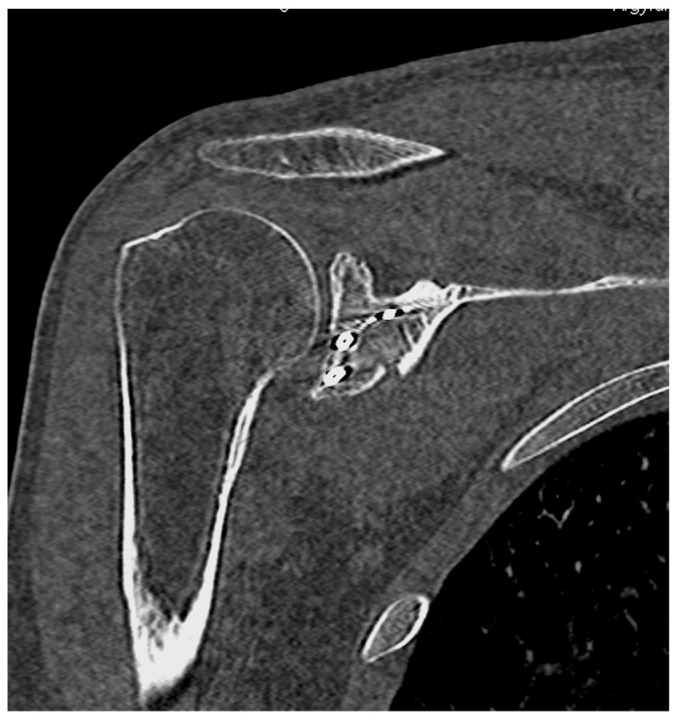
Postoperative CT scan showing fixation of the previous glenoid fossa fracture with two screws under arthroscopic visualization.

**Table 1 diagnostics-14-00908-t001:** Advantages and disadvantages of arthroscopy-assisted glenoid fossa fracture fixation.

Advantages	Disadvantages
Excellent visualization of articular surface	Technically challenging
Soft-tissue preservation	Not applicable in complex scapula fractures
Preservation of glenoid blood supply	No visualization of neurovascular structures
Reduced blood loss	Complications of arthroscopy
Improved cosmetics	
Decreased postoperative stiffness/weakness	
Earlier return to function	
Low incidence of infection	
Repair of concomitant intra-articular soft-tissue injuries	

**Table 2 diagnostics-14-00908-t002:** Demographic data and information regarding the applied fixation method of the included studies.

Author	Year	Type of Study	Number of Patients	Gender (M/F)	Mean Age (Years; Range)	Ideberg Classification	Fixation Method	Follow-Up (Months; Range)
Marsland [87]	2011	Case series	3	3/0	26.3 (17–40)	2 III, 1 IV	PCS	12
Bauer [10]	2006	Technical note	1	n/a	n/a	VI	Sutures	48
Gigante [100]	2003	Case report	1	0/1	25	VI	K-wires	6
Ulusoy [101]	2019	Case report	1	1/0	41	Vb	Cannulated screws	6
Zbili [103]	2017	Case report	1	0/1	22	III	Cannulated screw	12
Yang [104]	2011	Case series	18	12/6	41.2 (22–620	III	Cannulated screw	37.6 (24–61)
Qu [99]	2015	Case series	11	8/3	41 (27–55)	VI	PCS or sutures	21 (14–19)
Lin [30]	2022	Retrospective study	20	5/15	41.15 (25–61)	5 II, 4 III, 1 IV, 8 V, 2 VI	Cannulated or headless screws	>12
Bonczek [105]	2015	Case report	1	1/0	70	III	Cannulated screws	1.5

Abbreviations: F: female, M: male, n/a: not applicable, PCS: percutaneous cannulated screw.

**Table 3 diagnostics-14-00908-t003:** Details of postoperative shoulder range of motion at the last follow-up.

Author	Year	Flexion	Extension	Abduction	External Rotation	Internal Rotation
Bauer [10]	2006	170		150	60	T8
Gigante [100]	2003	normal	normal	normal	normal	normal
Ulusoy [101]	2019	170	45	140	45	L1
Zbili [103]	2017	170		160	40	T10
Yang [104]	2011	162.8 ± 8.3			67.2 ± 4.6	T8 ± 2
Qu [99]	2015	141.8 ± 18.9		160.9 ± 11.4	46.4 ± 6.4	
Lin [30]	2022	175.5 ± 10.9		168 ± 20.4	83.25 ± 7.6	T10 ± 2
Bonczek [105]	2015	170		130	30	L1

**Table 4 diagnostics-14-00908-t004:** Functional outcome of arthroscopically assisted glenoid fracture fixation.

Author	Year	VAS	ASES	Constant–Murley Score	UCLA Shoulder Score	DASH Score	Quick-DASH	Rowe Score
Marsland [87]	2011	n/a	n/a	95	n/a	n/a	n/a	n/a
Ulusoy [101]	2019	n/a	n/a	n/a	33	2	n/a	n/a
Zbili [103]	2017	n/a	n/a	100	35	n/a	0	n/a
Yang [104]	2011	0.7 ± 0.9	96 ± 5.4	96.8 ± 2.9	34.3 ± 0.9	n/a	n/a	n/a
Qu [99]	2015	n/a	87.3 ± 13.8	n/a	n/a	n/a	n/a	93.2 ± 11.2
Lin [30]	2022	1.5 ± 0.6	n/a	90.9 ± 9.2	n/a	6.8 ± 9.5	n/a	n/a

Abbreviations: ASES: American Shoulder and Elbow Surgeon Assessment, DASH: Disabilities of the Arm, Shoulder, and Hand, n/a: not applicable, UCLA: University of California at Los Angeles, VAS: Visual Analogue Scale.

## Data Availability

Not applicable.

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
