# Peer review of "The Role of Arthroscopy in Contemporary Glenoid Fossa Fracture Fixation"

_diagnostics, 2024, doi:10.3390/diagnostics14090908_

Round 1
Reviewer 1 Report
Comments and Suggestions for Authors
Thank you for allowing me to review this manuscript. I have throughoutfully read the paper and the topic is for sure interesting in my opinion but some sections need revision.
The information presented is new and the manuscript is appropriate for the journal in my opinion.
Title and abstract is ok.
In the introduction section, conservative treatment indications, surgical approach and surgical techniques are discussed in excessive and unnecessary detail. This causes repetition and unnecessary length in the text. Briefly stating what the treatment options are and mentioning the place of the arthroscopic technique will be sufficient for the introduction part.
The inclusion criteria of the studies referenced in this article, which is stated to be a systematic review study, should be mentioned.
If possible, giving examples of intraoperative images/preoperative-postoperative radiological imaging for fixation with arthroscopic technique in the management section may enrich the text.
I would like to congratulate the authors on the use of current literature.
Author Response
Dear Editor,
We would like to thank you for accepting to reconsider our manuscript titled: “The Role of Arthroscopy in Contemporary Glenoid Fossa Fracture Fixation” for publication in the Diagnostics journal.
We would also like to thank the reviewers for their insightful comments. All raised points have been addressed and the manuscript has been revised according to their suggestions. All text changes in the manuscript have been highlighted. For reviewing purposes, the comments have been addressed one by one.
In more detail:
Reviewer #1
Comment: “In the introduction section, conservative treatment indications, surgical approach and surgical techniques are discussed in excessive and unnecessary detail. This causes repetition and unnecessary length in the text. Briefly stating what the treatment options are and mentioning the place of the arthroscopic technique will be sufficient for the introduction part.”
Reply: Thank you for your comment. The introduction section has been revised according to your suggestion.
Comment: “The inclusion criteria of the studies referenced in this article, which is stated to be a systematic review study, should be mentioned.”
Reply: Thank you for the comment. The inclusion criteria of the review are clearly presented in Lines 290-293.
Comment: “If possible, giving examples of intraoperative images/preoperative-postoperative radiological imaging for fixation with arthroscopic technique in the management section may enrich the text.”
Reply: Thank you for the comment. Apart from the arthroscopic view of a glenoid fossa fracture, we present the corresponding CT view after fixation (Figure 4).
Reviewer 2 Report
Comments and Suggestions for Authors
Authors requested to add some figures under heading of images and comparative table of effectiveness of arthroscopic interventions in previously published literatures in discussion section
Author Response
Dear Editor,
We would like to thank you for accepting to reconsider our manuscript titled: “The Role of Arthroscopy in Contemporary Glenoid Fossa Fracture Fixation” for publication in the Diagnostics journal.
We would also like to thank the reviewers for their insightful comments. All raised points have been addressed and the manuscript has been revised according to their suggestions. All text changes in the manuscript have been highlighted. For reviewing purposes, the comments have been addressed one by one.
In more detail:
Reviewer # 2
Comment: “Authors requested to add some figures under heading of images and comparative table of effectiveness of arthroscopic interventions in previously published literatures in discussion section.”
Reply: Thank you for your comment. We added three tables in section six of the article (“The role of arthroscopy”) with information about the demographic data, applied surgical technique, range of motion and functional outcomes at the last follow-up. In addition, a CT view of a united glenoid fossa fracture that treated with arthroscopically assisted internal fixation is presented (Figure 4).